# The beneficial effect of chronic muscular exercise on muscle fragility is increased by *Prox1* gene transfer in dystrophic *mdx* muscle

**Alexandra Monceau**[1], **Clément Delacroix**[1], **Mégane Lemaitre**[2], **Gaelle Revet**[3], **Denis Furling**[1], **Onnik Agbulut**[3], **Arnaud Klein**[1], **Arnaud Ferry**[1,4]*

1 Myology Center of Research, Association of Myology Institute, UMRS974, Inserm, Sorbonne Université, Paris, France, 2 UMS 28, Sorbonne Université, Paris, France, 3 Biological Adaptation and Ageing, Institute of Paris-Seine Biology, UMR CNRS 8256, Inserm ERL U1164, Sorbonne Université, Paris, France, 4 Université Paris Cité, Paris, France

* arnaud.ferry@upmc.fr

**Data Availability Statement:** All relevant data are within the manuscript and its Supporting Information files.

## Abstract

### Purpose

Greater muscle fragility is thought to cause the exhaustion of the muscle stem cells during successive degeneration/repair cycles, leading to muscle wasting and weakness in Duchenne muscular dystrophy. Chronic voluntary exercise can partially reduce the susceptibility to contraction induced-muscle damage, i.e., muscle fragility, as shown by a reduced immediate maximal force drop following lengthening contractions, in the dystrophic *mdx* mice. Here, we studied the effect of Prospero-related homeobox factor 1 gene (*Prox1*) transfer (overexpression) using an AAV on fragility in chronically exercised *mdx* mice, because *Prox1* promotes slower type fibres in healthy mice and slower fibres are less fragile in *mdx* muscle.

### Methods

Both *tibialis anterior* muscles of the same *mdx* mouse received the transfer of *Prox1* and PBS and the mice performed voluntary running into a wheel during 1 month. We also performed *Prox1* transfer in sedentary *mdx* mice. *In situ* maximal force production of the muscle in response to nerve stimulation was assessed before, during and after 10 lengthening contractions. Molecular muscle parameters were also evaluated.

### Results

Interestingly, *Prox1* transfer reduced the isometric force drop following lengthening contractions in exercised *mdx* mice (p < 0.05 to 0.01), but not in sedentary *mdx* mice. It also increased the muscle expression of *Myh7* (p < 0.001), MHC-2x (p < 0.01) and *Trpc1* (p < 0.01), whereas it reduced that one of *Myh4* (p < 0.001) and MHC-2b (p < 0.01) in exercised *mdx* mice. Moreover, *Prox1* transfer decreased the absolute maximal isometric force (p < 0.01), but not the specific maximal isometric force, before lengthening contraction in exercised (p < 0.01) and sedentary *mdx* mice.

**Funding:** The author(s) received no specific funding for this work.

**Competing interests:** The authors have declared that no competing interests exist.

## Conclusion

Our results indicate that Prox1 transfer increased the beneficial effect of chronic exercise on muscle fragility in *mdx* mice, but reduced absolute maximal force. Thus, the potential clinical benefit of the transfer of Prox1 into exercised dystrophic muscle can merit further investigation.

## Introduction

Duchenne muscular dystrophy (DMD), the most common X-linked inherited muscular disease, is caused by mutations in the *DMD* gene, leading to dystrophin deficiency that results in skeletal muscle fibre injury and progressive muscle wasting and weakness. Dystrophin is a costameric protein that plays a role in force transmission and sarcolemma stability in skeletal muscle [1]. In line, muscle of dystrophin-deficient *mdx* mouse, the "classic" animal model for DMD, exhibits two important functional dystrophic features. First, muscular weakness that is the decrease of the specific maximal force (the absolute maximal force generated relatively to muscle cross-sectional area or weight) with an unmodified/maintained absolute maximal force due to the muscle hypertrophy [2]. A second robust phenotype is muscle fragility that is revealed by the high susceptibility of the fast and low oxidative *mdx* muscle for damage caused by the lengthening (eccentric) contractions, leading in particular to an immediate marked force drop following lengthening contractions [2–4]. This force drop is proportional to both the length of the stretch and the absolute maximal lengthening force produced during the first contraction in fast *mdx* muscle [3, 4]. It was also found no muscle histological structural change immediately following lengthening contractions in *mdx* mice [5], as well as no reduction in maximal force of permeabilized muscle fiber [5, 6]. The greater fragility in *mdx* mice is associated to reduced muscle excitability [5, 7–9], and several genes coding ion membrane channels interacting with dystrophin are involved in muscle excitability, such as *Scn4a*, *Cacna1s*, *Slc8a1*, *Trpc1* and *chrna1* [10, 11]. Increased fragility is also related to NADPH oxidase 2 (NOX2) activity [12–14] and aggravated by inactivation of *Utrn* and *Des* coding utrophin and desmin respectively in *mdx* mice [15, 16].

The fragility of the dystrophic muscle is thought to cause the exhaustion of the muscle stem cells during successive degeneration/repair cycles [17]. Thus, attempt to reduce this fragility is very important because it has the potential to slow the progression of the dystrophic disease. Interestingly, chronic muscular exercise can improve (reduced) the fragility in *mdx* mice [9, 18–20]. In particular, voluntary running decreases fragility, i.e., reduces the force drop following lengthening contractions, in *mdx* mouse fast muscle [9, 19], whereas physical inactivity aggravates it [19]. Recently, it was found that the reduced fragility induced by voluntary running in *mdx* mice was related to calcineurin pathway activation, and changes in the program of genes involved in slower contractile features of muscle fibre and genes coding membrane ions channels involved in muscle excitability [9]. However, voluntary running only partly reduced the susceptibility to exercise-induced muscle damage [9, 19], so it would be interesting to combined the effects of exercise with those of another treatment.

While voluntary exercise offers potential therapeutic benefit, additional adjunct therapies could further improve functional dystrophic features. In the recent years, genetic or pharmacological treatments promoting slower and more oxidative fibres are been shown to be beneficial in the *mdx* mice. In fact, several studies support the idea that activation of the AMPK,

calcineurin, E2F1, ERRγ, IGF1, SIRT1 and PGC1 signalling pathways alleviates some of the dystrophic features in *mdx* muscle [21–30]. For example, genetic activation of calcineurin pathway improves fragility in fast muscle of the *mdx* mouse, but decreases maximal force production, thus, aggravating weakness [28]. Recently, in healthy fast muscle, it was demonstrated that the loss of Prospero-related homeobox factor 1 (*Prox1*), a transcription factor essential for the development of several organs like lymphatic vessels and highly conserved among vertebrates, represses the expression of slow contractile genes, whereas its overexpression via *Prox1* transfer has the opposite effect and down-regulates the fast contractile genes, [31, 32]. In particular, the inactivation of *Prox1* reduces the expression of the slowest myosin heavy chain *Myh7* in fast healthy muscle, without affecting oxidative capacity (succinate dehydrogenase staining) and absolute maximal force [32]. *Prox1*, that is more expressed in slow fibres, is involved in the activation of the NFAT/calcineurin pathway, and promotes the slow contractile gene program in healthy muscle [31].

The principal purpose of the present study was to determine whether *Prox1* transfer using an adeno-associated vectors (AAV9) carrying the Prox1 construct reduced muscle fragility in voluntary exercised *mdx* mice, with fragility being defined as the immediate loss of muscle function (i.e., maximal force drop) following lengthening contractions. The study including physiological outcome measurement of fragility was complemented by molecular analyses. Because we found that voluntary running and *Prox1* transfer have additive beneficial effects on fragility, a second set of experiment was performed to compare the effect of *Prox1* transfer on fragility between voluntary exercised and sedentary *mdx* mice. Interestingly, *Prox1* transfer did not reduced fragility in sedentary *mdx* mice.

## Materials and methods

### Animal groups and voluntary running

All procedures were performed in accordance with national and European legislations and were approved by our institutional Ethics Committee "Charles Darwin" (Project # 01362.02). Male mice with exon 23 mutation in the *dmd* gene encoding dystrophin (Mdx mice) were used (hybrid background C57Bl/6 x C57Bl/10). Mice (2–3 months of age) were randomly divided into different control and experimental groups (Fig 1). In the first set of experiment, Mdx mice were placed (Mdx+W) in separate cages containing a wheel and were allowed to run 1-month ad libitum. The muscles of Mdx mouse runners received (Mdx+W+P) or not (Mdx+W) *Prox1* transfer into the muscle 3 days before the initiation of voluntary exercise. The running distances were collected and daily running distance was $4.2 \pm 0.1$ km/day. A group of sedentary Mdx mice was also studied (Mdx). At the end of the experiment, the body weight of the Mdx+W+P/Mdx+W mice and Mdx mice was $29.9 \pm 0.3$ g and $31.4 \pm 0.2$ g respectively (p = 0.019). The first set of experiment was performed to study the effect of *Prox1* transfer in exercised Mdx mice. Because we found an effect of *Prox1* transfer on fragility in Mdx+W+P muscle, we then performed a second set of experiment to compare the effect of *Prox1* transfer on fragility between voluntary exercised muscle and sedentary muscle. In the second set of experiment, the muscles of sedentary Mdx mice received (Mdx+P) or not (Mdx) *Prox1* transfer. The muscles were measured and collected 4 weeks after *Prox1* transfer.

### *Prox1* transfer

To overexpress Prox1, adeno-associated vectors (AAV9) carrying the *Prox1* construct (AAV--Prox1) [31] was injected in one of the Tibialis anterior (TA) muscles of the mouse ($2.1 \times 10^{11}$ vector genomes). The other TA muscle (control muscle) of the same mouse was injected with

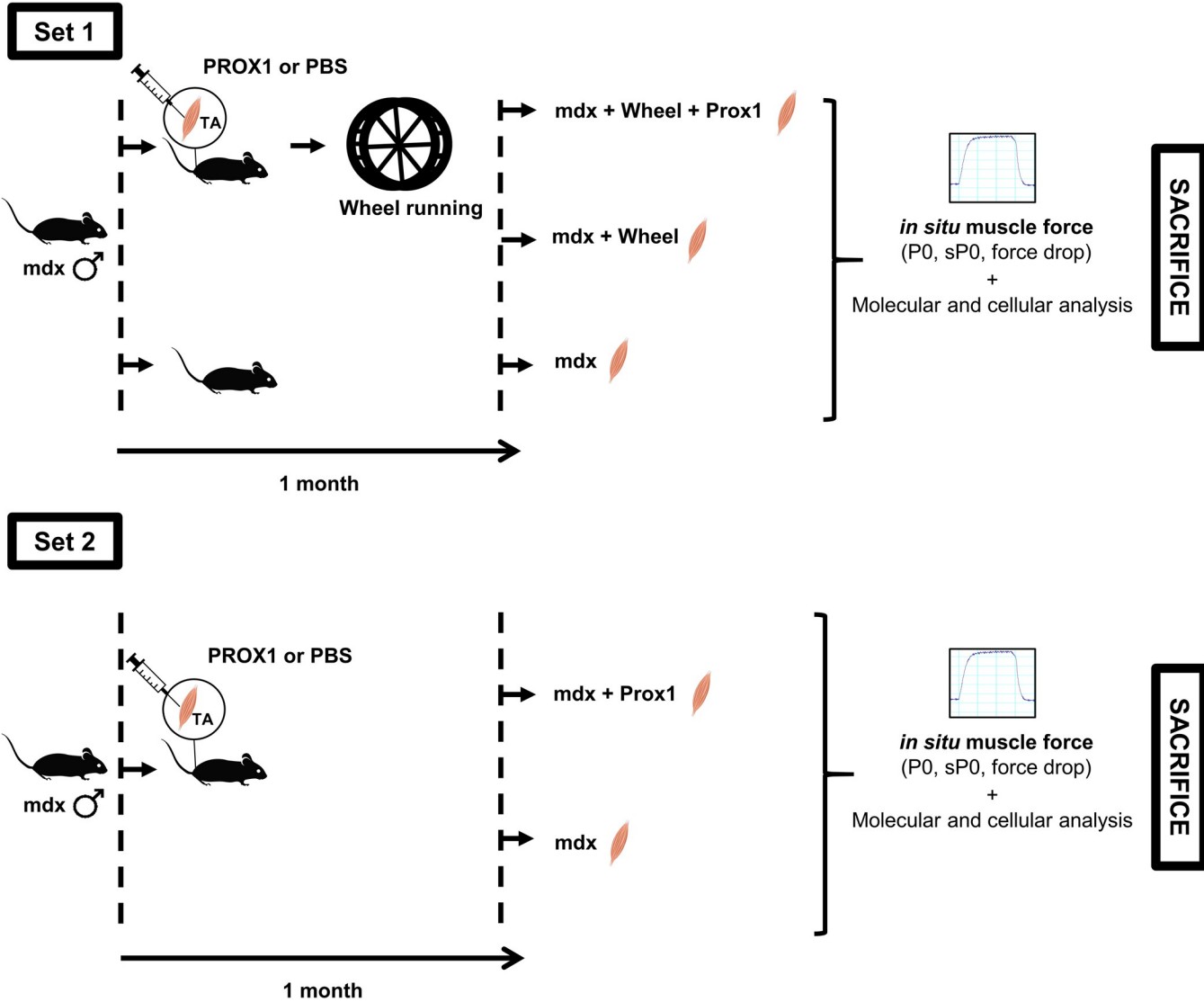

**Fig 1. Experimental design.** Two sets of experiments were performed. In the first set of experiment, we want to determine the effect of *Prox1* transfer on fragility in voluntary exercised mdx mice. AAV-Prox1 and PBS were injected in TA muscles of the same *mdx* mice before voluntary exercise. The aim of the second set of experiment was to compare the effect of *Prox1* transfer between voluntary exercised and sedentary mdx mice.

saline solution only (Fig 1). The mouse was anesthetized (3% isoflurane) and TA muscles were injected (30 μl). Briefly, hProx1 cDNA's were cloned into psub plasmid (promoter CMV) [31]. The plasmid was purified using the PureYield™ endotoxin-free Plasmid Maxiprep System (Promega, Lyon, France) and then verified by restriction enzyme digestion and by sequencing (Eurofins MWF Operon, Ebersberg, Germany). The AAV-Prox1 was produced in human embryonic kidney 293 cells by the triple-transfection method using the calcium phosphate precipitation technique. The virus was then purified by 2 cycles of cesium chloride gradient centrifugation and concentrated by dialysis. The final viral preparations were kept in PBS solution at -80˚C. The number of viral genomes was determined by a quantitative PCR. Titer for AAV-Prox1 was $7.1 \times 10^{12}$ vector genomes (vg).ml$^{-1}$.

## Muscle fragility measurement

Muscle fragility (susceptibility to contraction induced damage) was evaluated by measuring the *in situ* TA muscle contraction properties in response to nerve stimulation, as described previously [5]. Fragility was estimated from the isometric force drop resulting from lengthening contraction-induced damage. Briefly, mice were anesthetized using pentobarbital (60 mg/kg, ip). Body temperature was maintained at 37˚C using radiant heat. The knee and foot were fixed with pins and clamps and the distal tendon of the muscle was attached to a lever arm of a servomotor system (305B, Dual-Mode Lever, Aurora Scientific) using a silk ligature. The sciatic nerve was proximally crushed and distally stimulated by a bipolar silver electrode using supramaximal square wave pulses of 0.1 ms duration. We first determined the optimal length (L0, length at which maximal isometric force was obtained during the tetanus). Once L0 was obtained, a maximal isometric contraction of the TA muscle was initiated during the first 500 ms. Then, muscle lengthening (10% L0) at a velocity of 5.5 mm/s (0.85 fibre length/s) was imposed during the last 200 ms. Nine lengthening contractions of the TA muscles were performed in Mdx mice, each separated by a 60 s rest period. Absolute maximal isometric force was measured 1 min after each lengthening contraction and expressed as a percentage of the initial maximal force (force drop). Absolute maximal isometric force measured before the first lengthening contraction was also normalized to the muscle mass in order to calculate the specific maximal isometric force, an index of muscle weakness. In addition, we measured the absolute maximal lengthening force during the first lengthening contraction, and index of the muscle stress. After contractile measurements, the animals were killed with cervical dislocation.

## Real-time quantitative PCR (polymerase chain reaction)

Muscles (TA) were snap frozen in liquid nitrogen and stored at −80˚C until use. Total RNA was isolated from TA muscles using Trizol (Invitrogen). Complementary DNA (cDNA) was then synthesized from 1 µg of total RNA using the RevertAid First Strand cDNA Synthesis kit with random hexamers, according to the manufacturer's instructions (Thermo Scientific). RT-PCR was performed on a LightCycler 480 System at the platform iGenSeq of the Institut du Cerveau et de la Moelle epinière, using LightCycler 480 SYBR Green I Master Mix (Roche, Basel, Switzerland) [5]. The expression of *Hmbs* was used as reference transcript because it's expression did not differ between groups. The 2-ΔΔCP method has been used as a relative quantification strategy for quantitative real-time polymerase chain reaction (qPCR) data analysis. All sequences of primers used are presented in Table 1.

## SDS-PAGE electrophoresis of MHC isoforms (proteins)

The muscles were extracted on ice for 60 min in four volumes of extracting buffer containing 0.3 M NaCl, 0.1 M NaH2PO4, 0.05 M Na2HPO4, 0.01 M Na4P2O7, 1 mM MgCl2, 10 mM EDTA, and 1.4 mM 2-mercaptoethanol (pH 6.5). Following centrifugation, the supernatants were diluted 1:1 (vol/vol) with glycerol and stored at -20˚C. MHC isoforms (proteins) were separated on 8% polyacrylamide gels, which were made in the Bio-Rad mini-Protean II Dual slab cell system. The gels were run for 31 h at a constant voltage of 72 V at 4˚C [33]. Following migration, the gels were silver stained. The gels were scanned using a video acquisition system. The relative level of MHC isoforms was determined by densitometric analysis using Image J software.

## Histology

Transverse serial sections (8 µm) of TA muscles were obtained using a cryostat, in the mid-belly region. For determination of muscle fibre diameter (min ferret), frozen unfixed sections

**Table 1. Sequences of primers used.**

| Gene | Forward | Reverse |
|---|---|---|
| **House keeping gene** | | |
| Hmbs | 5'– AGGTCCCTGTTCAGCAAGAA –3' | 5'– TGGGCTCCTCTTGGAATGTT –3' |
| **Genes of interest** | | |
| Cacna1 | 5'–CCTCATCAGCAAGAAGCAGG–3' | 5'–TATGACAGACAGACCCTGGC–3' |
| Chrna1 | 5'– TGGTCTTTGTCATTGCGTCC –3' | 5'– GATAAAAACCTTCCGCACCCA –3' |
| Des | 5'– GTCCTCACTGCCTCCTGAAG–3' | 5'– AGCATGAAGACCACAAAGGG–3' |
| Fn14 | 5'– AGGGGCTATAATGCCACTCC –3' | 5'– GGGAGATGGTTGTTTCCGTG –3' |
| Fst | 5'– CGAGTGTGCCATGAAGGAAG –3' | 5'– GGTCTTCCTCCTCCTCCTCT –3' |
| Gadd45 | 5'– GGTGACGAACCCACATTCAT –3' | 5'– GATTAATCACGGGCACCCAC –3' |
| Gp91phox | 5'–TCACATCCTCTACCAAAACC–3' | 5'–CCTTTATTTTTCCCCATTCT–3' |
| Hdac4 | 5'– AAGTAGCTGAGAGACGGAGC –3' | 5'– GCATGCGGAGTCTGTAACAT –3' |
| Igf1 | 5'–ACAAGCCCACAGGCTATGGCTC –3' | 5'–AGTCTCCTCAGATCACAGCTCCG –3' |
| Lc3 Map1lc3a | 5'– CATGAGCGAGTTGGTCAAGA –3' | 5'– CCATGCTGTGCTGGTTGA –3' |
| Mafbox | 5'–TCACAGCTCACATCCCTGAG–3' | 5'– TCAGCCTCTGCATGATGTTC–3' |
| Mstn | 5'– GCTACCACGGAAACAATCAT–3' | 5'–CAATACTCTGCCAAATACCA–3' |
| Murf1 | 5'–TGAGGTGCCTACTTGCTCCT–3' | 5'–GTGGACTTTTCCAGCTGCTC–3' |
| Myh2 | 5'–AAGCGAAGAGTAAGGCTGTC–3' | 5'–GTGATTGCTTGCAAAGGAAC–3' |
| Myh4 | 5'–ACAAGCTGCGGGTGAAGAGC–3' | 5'–CAGGACAGTGACAAAGAACG–3' |
| Myh7 | 5'–AGGTGTGCTCTCCAGAATGG–3' | 5'–CAGCGGCTTGATCTTGAAGT–3' |
| P47phox | 5'–AGAACAGAGTCATCCCACAC–3' | 5'–GCTACGTTATTCTTGCCATC–3' |
| PrxII | 5'–GGTTTGGGCCACGCATAAAA–3' | 5'–GCCATGACTGCGTGAGCAAG–3' |
| Prox1 | 5'–GCTACCCCAGCTCCAACATGCT–3' | 5'–TGATGGCTTGACGCGCATACTTCT–3' |
| Rac1 | 5'–GTAAAACCTGCCTGCTCATCA–3' | 5'–GAGAGGGGACGCAATCTGT–3' |
| Redd1 Ddit4 | 5'– ACTACTGACCTGTTCGAGGC –3' | 5'– TCAAGTGTCGAAGATCCCGA –3' |
| Redd2 Ddit4l | 5'– GTGCAGCCCCATCAAAACATA –3' | 5'– GAAGCCATGCTCTTGTCACTG –3' |
| Sdha | 5'–TTACAAAGTGCGGGTCGATG–3' | 5'–GTGTGCTTCCTCCAGTGTTC–3' |
| Scn4a | 5'–GCAACCTGGTGGTCCTGAAT–3' | 5'–CAGCCCCAAGAGGAAGGTTT–3' |
| Slc8a1 | 5'– GGAGACTGCTCGTGTGTCTA –3' | 5'– TGTTGGTTGGCCTGAGAGAT –3' |
| Smox | 5'–AAGTTGTGAATCCAGTGGCG–3' | 5'–GTCTCCAAGCCTCACACTCT–3' |
| Tnni1 | 5'–ATGGAGGAGGTGGATCTGC–3' | 5'–TTCAAATTTGGCCCGGCAC–3' |
| Trpc1 | 5'– TCTATAGATGTCTGGCCAGTCC–3' | 5'– CATTTTGCACTGACGGGCTA–3' |
| Utrn | 5'– CACTATGACCCCTCCCAGTC –3' | 5'– CGCTTCCTGTTGTAGAGCTG –3' |

were blocked 1h in phosphate buffer saline plus 2% bovine serum albumin, 2% fetal bovine serum. Sections were then incubated overnight with primary antibodies against laminin (Sigma, France). After washes in PBS, sections were incubated 1 h with secondary antibody (Alexa Fluor, Invitrogen). Slides were finally mounted in Fluoromont (Southern Biotech). Images were captured using a digital camera (Hamamatsu ORCA-AG) attached to a motorized fluorescence microscope (Zeiss AxioImager.Z1), and morphometric analyses were made using the software ImageJ. We attempt to analyze all the fibers of the muscle section, but some were excluded from the analysis for reasons of improper labeling (mean: 1474 fibres measured per muscle).

## Statistical analysis

Groups were statistically compared using the Prism software v8 (GraphPad, La Jolla, CA, USA). Data were tested for homogeneity of variance using a Brown-Forsythe test. For the first

set of experiment, one-way ANOVA was used to analyze the following variables: mRNA expression, absolute and specific maximal force, absolute maximal lengthening force, the ratio of absolute maximal lengthening force to the absolute maximal lengthening force, and muscle weight. Fragility was analyzed by two-way ANOVA, groupes (Mdx, Mdx+W, Mdx+W+P) by lengthening contractions (0, 3, 6, 9), with the repeated measures on lengthening contractions. Unpaired t-test with Welch's correction was used to analyze the % of MHC-2x and MHC-2a (electrophoresis) and body weight of the mice. For experiment 2, unpaired t-test with Welch's correction was used for the following variable: mRNA expression, absolute and specific maximal force, absolute maximal lengthening force, and muscle weight. Fragility was analyzed by two-way ANOVA, groupes (Mdx, Mdx+P) by lengthening contractions (0, 3, 6, 9), with the repeated measures on lengthening contractions. Moreover, when significant main effect (ANOVA) was observed, multiple-comparisons were performed with Tukeys test. Finally, when significant interaction was found (ANOVA), differences were tested with Holm-Sidak test. Values are means ± SEM.

## Results

### *Prox1* transfer in voluntary exercised Mdx muscle promotes slower contractile features

In the first set of experiment, we first determined whether *Prox1* transfer increased slower contractile features in voluntary exercised Mdx mice. *Prox1* transfer into the TA muscle markedly increased the expression of *Prox1* (x 37.0) in voluntary exercised Mdx TA muscle (Mdx+W +P) as compared to voluntary exercised Mdx TA muscle (Mdx+W)($p < 0.0001$) (Fig 2A), as assessed by qPCR analysis. We also found that the expression of *Myh7* coding for MHC-1 (x 15.1)($p < 0.001$) was increased in Mdx+W+P muscle as compared to Mdx+W muscle, whereas that of *Myh4* coding for MHC-2b was reduced (x 0.6) (Fig 2B) ($p < 0.001$). In agreement, using gel electrophoresis technique, we found that the relative amounts (percentage of total) of MHC-2b protein were reduced (x 0.8, $p < 0.01$) whereas that of MHC-2x protein was increased (x 1.6, $p < 0.01$), respectively (Fig 2C) in Mdx+W+P muscle as compared to Mdx +W muscle. In contrast, there was no difference between Mdx+W+P and Mdx+W muscles in the expression of a marker of oxidative capacity, *Sdha*, a gene encoding a complex of the mitochondrial respiratory chain (Fig 2B).

These data indicate that intramuscular delivery of AAV-*Prox1* induced a fast to slow contractile transition in the TA muscle of voluntary exercised Mdx mice.

### *Prox1* transfer in voluntary exercised Mdx muscle further improves muscle fragility

The first set of experiment revealed that the immediate isometric force drop following lengthening contractions in Mdx+W muscle was reduced as compared to Mdx muscle ($p < 0.0001$) (Fig 3A). Interestingly, *Prox1* transfer in voluntary exercised Mdx muscle further reduced the isometric force drop following lengthening contractions (Fig 3A). In fact, the isometric force drops following the 6th ($p < 0.05$) and 9th ($p < 0.01$) lengthening contractions were lower in Mdx+W+P muscle as compared to Mdx+W muscle (Fig 3A), indicating that *Prox1* transfer improved (reduced) fragility in voluntary exercised Mdx muscle.

The fast to slower contractile conversion described above can explained, at least in part, the improved fragility in Mdx+W+P muscle. Moreover, we tested the possibility that *Prox1* transfer also improved fragility via the modifications of the expression of genes coding membrane ions channels. The expression of *Trpc1* encoding for transient receptor potential cation

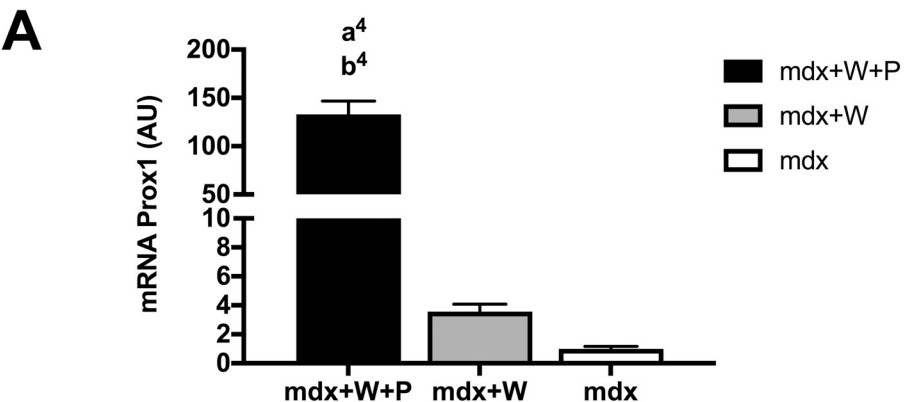

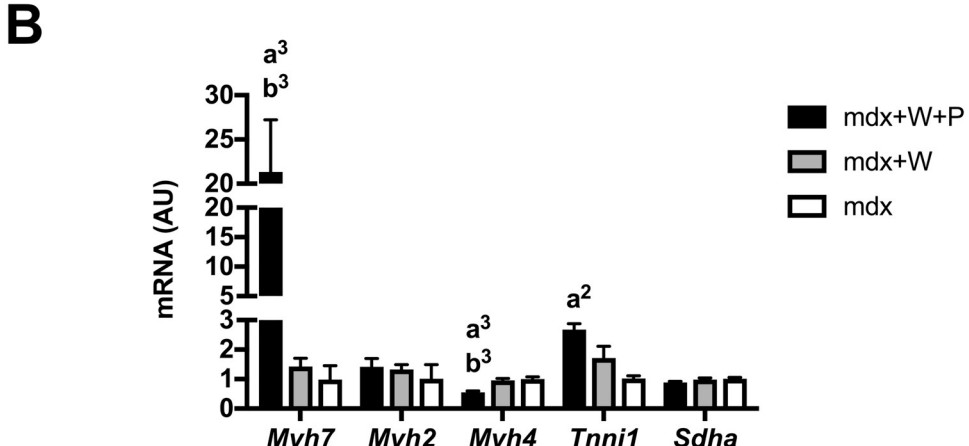

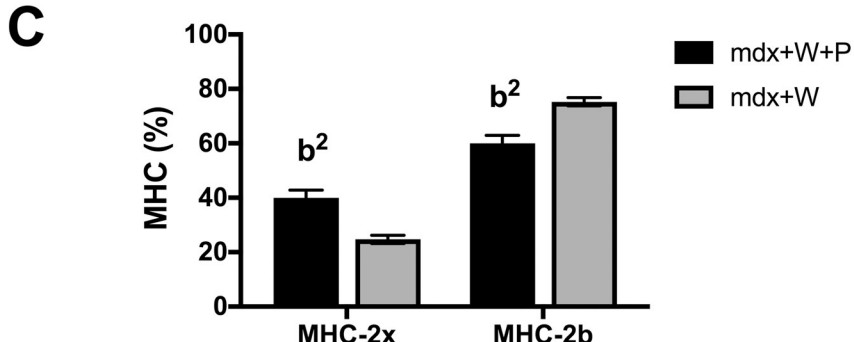

**Fig 2. Effect of *Prox1* transfer on the expression of *Prox1* and markers of fibre type specification in voluntary exercised mdx mice (first set of experiment).** (A) *Prox1* expression in Mdx+W+P and Mdx+P muscle. N = 6–8 per group. (B) Expression of genes encoding fibre type specific contractile proteins in Mdx+W+P and Mdx+P muscle. N = 6–8 per group. (C) Relative amounts of MHC-2x and MHC-2b proteins in Mdx+W+P and Mdx+P muscle. N = 3 per group. Mdx+W+P: voluntary exercised mdx muscle that received Prox1 transfer into the muscle. Mdx+W: voluntary exercised mdx muscle. Mdx: mdx muscle. a2, a3, a4: significant different from Mdx, $p < 0.01$, $p < 0.001$, $p < 0.0001$, respectively. b2, b3, b4: significant different from Mdx+W, $p < 0.01$, $p < 0.001$, $p < 0.0001$, respectively.

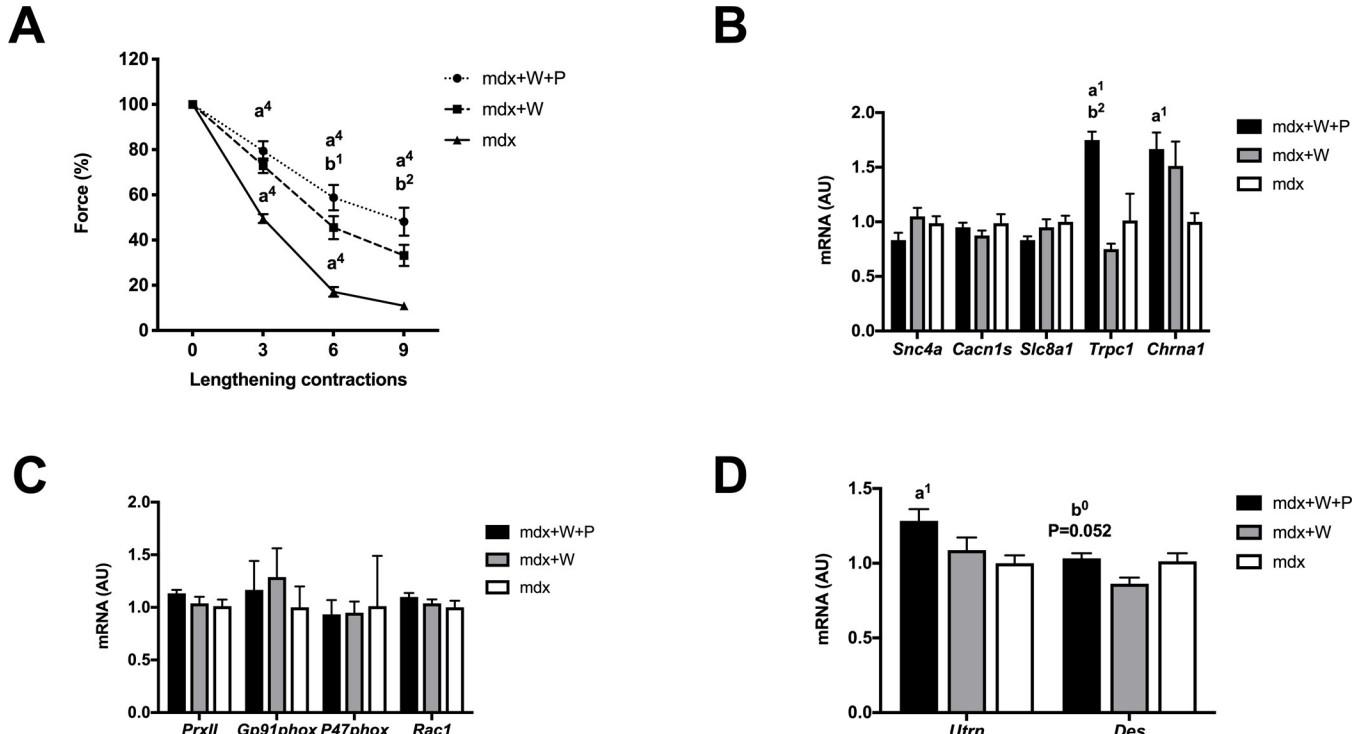

**Fig 3. Effect of *Prox1* transfer on fragility (susceptibility to contraction induced-muscle damage) and related gene expression in voluntary exercised mdx mice (first set of experiment).** (A) Force drop following lengthening contractions (Fragility) in Mdx+W+P and Mdx+P muscle. n = 6–8 per group. (B) Expression of genes encoding ion channels, related to excitability in Mdx+W+P and Mdx+P muscle. N = 6–8 per group. (C) Expression of genes, related to NADPH oxidase 2 (NOX2) in Mdx+W+P and Mdx+P muscle. N = 6–8 per group. (D) Expression of genes encoding utrophin (*Utrn*) and desmin (*Des*) in Mdx+W+P and Mdx+P muscle. N = 6–8 per group. Mdx+W+P: voluntary exercised mdx muscle that received Prox1 transfer into the muscle. Mdx+W: voluntary exercised mdx muscle. Mdx: mdx muscle. a1, a4: significant different from Mdx, $p < 0.05$, $p < 0.0001$, respectively. b1, b2: significant different from Mdx+W, $p < 0.05$, $p < 0.01$, respectively.

channel subfamily C member 1 (x 2.1) was increased in Mdx+W+P muscle as compared to Mdx+W muscle ($p < 0.01$) (Fig 3B). No difference between Mdx+W+P and Mdx+W muscles was observed concerning the expression of *Scn4a, Cacna1s, Slc8a1* and *Chrna1* (Fig 3B). Then, we determined whether the reduced isometric force drop following lengthening contractions induced by *Prox1* transfer was associated to change (decrease) in NOX2 pathway. We found no change in the expression of *PrxII, Gp91phox, P47phox* and *Rac1* (Fig 3C) in Mdx+W+P muscle as compared to Mdx+W muscle (Fig 3C). We also determined whether *Prox1* transfer increased *Utrn* and *Des* expression. The expression of *Utrn* was not increased in Mdx+W+P muscle as compared to Mdx+W muscle, whereas that one of *Des* increased (x 1.2) in Mdx+W muscle, although not significantly ($p = 0.052$) (Fig 3D).

Thus, the improved TA muscle fragility induced by *Prox1* transfer in voluntary exercised mice was associated with the modification of expression of MHC-2b and MHC-2x proteins and several genes involved in different aspects of muscle function and structure (*Myh7, Myh4, Trpc1*).

### *Prox1* transfer in voluntary exercised Mdx muscle reduced absolute isometric maximal force

In addition, the first set of experiment revealed that *Prox1* transfer combined to voluntary running and voluntary running alone did not affect specific maximal isometric force before

lengthening contractions (Fig 4A). However, absolute maximal isometric force was reduced in Mdx+W+P muscle (x 0.6) as compared to Mdx+W muscle ($p < 0.01$) (Fig 4B). Similarly, absolute maximal lengthening force was lower (x 0.6) in Mdx+W+P muscle (157.2 g ± 7.5) compared to Mdx+W muscle (240.0 g ± 10.8) muscle ($p < 0.01$). In addition, the ratio of absolute maximal lengthening force to the absolute maximal isometric force was not different between Mdx+W+P muscle (1.9 ± 0.1) and Mdx+W muscle (1.8 ± 0.1).

The reduced absolute maximal isometric force was related to a lower muscle weight (x 0.7)($p < 0.001$) (Fig 4C) and reduced fibre diameters ($p < 0.01$) (Fig 4D and 4E) in Mdx +W+P muscle as compared to Mdx+W muscle. Numerous genes encoding proteins are involved in muscle atrophy, growth and maintenance [34, 35]. The ubiquitin-proteasome system plays a key role in triggering muscle atrophy when the expressions of *Murf1* and *Mafbox* are increased. Quantitative real-time PCR revealed that the expressions of these genes were not increased in Mdx+W+P muscle as compared to Mdx+W muscle (Fig 4F). We then analyzed another atrophic mechanism, autophagy, which involves a battery of genes including *Lc3* which could contribute to the degradation of muscle proteins [36]. We did not find any change in *Lc3* expression in Mdx+W+P muscle (Fig 4F). Similarly, *Gadd45*, *Hdac4*, *Fn14*, *Redd1*, *Redd2*, *Mstn*, *Fst*, *Igf1*, and *Smox* genes also did not seem to participate to the atrophic state of Mdx+W+P muscle (Fig 4F). For example, *Mstn*, the negative regulator of muscle growth, was down-regulated in Mdx+W+P muscle as compared to Mdx+W muscle ($p < 0.001$) (Fig 4F).

Thus, the reduction in maximal isometric force induced by *Prox1* transfer in voluntary exercised Mdx muscle was related to decreased muscle weight and increased expression of *Mstn*.

### *Prox1* transfer in sedentary Mdx muscle promotes slower contractile features but does not reduce muscle fragility

A second set of experiment was performed to compare the effect of *Prox1* transfer on fragility between voluntary Mdx mice and sedentary Mdx mice. Similarly to voluntary exercised Mdx muscle, *Prox1* transfer in sedentary Mdx muscle (Mdx+P muscle) increased the expressions of *Prox1* (x 27.3)($p < 0.0001$) (Fig 5A), *Myh7* (x 6.2)($p < 0.05$) (Fig 5B), and reduced that one of *Myh4* (x 0.7)($p < 0.05$) (Fig 5B) compared to sedentary Mdx muscle. In contrast to voluntary exercised Mdx muscle, *Prox1* transfer increased the expression of *Tnni1* (x 2.4)($p < 0.01$), reduced the expression of *Sdha* (x 0.8) (Fig 5B) ($p < 0.01$) and did not change the relative amounts of MHC-2b and MHC-2x proteins (Fig 5C) in Mdx+P muscle as compared to Mdx muscle. Overall, intramuscular delivery of AAV-*Prox1* also induced a fast to slow contractile conversion in the TA muscle of sedentary Mdx mice, but without consequence at the MHC protein level.

In contrast to voluntary exercised Mdx muscle, we found in the second set of experiment that the isometric force drop following lengthening contractions was not significantly reduced by *Prox1* transfer in sedentary Mdx muscle because there was no significant difference between Mdx+P muscle and Mdx muscle (Fig 6A). Similarly to voluntary exercised Mdx muscle, *Prox1* transfer in Mdx+P muscle increased *Trpc1* expression ($p < 0.01$) (Fig 6B), but to lesser extent (x 1.4), did not alter the expression of *PrxII*, *Gp91phox*, *P47phox* and *Rac1* (Fig 6C), and increased not significantly *Des* expression ($p = 0.059$) (Fig 6D). In contrast to voluntary exercised Mdx muscle, the expression of *Cacn1s* and *Chrna1* was increased in Mdx+P muscle compared to Mdx muscle ($p < 0.01$) (Fig 6B).

Thus, *Prox1* transfer in sedentary Mdx muscle does not reduced fragility, did not change the expression of MHC-2b and MHC-2x, whereas it altered the expression of several genes

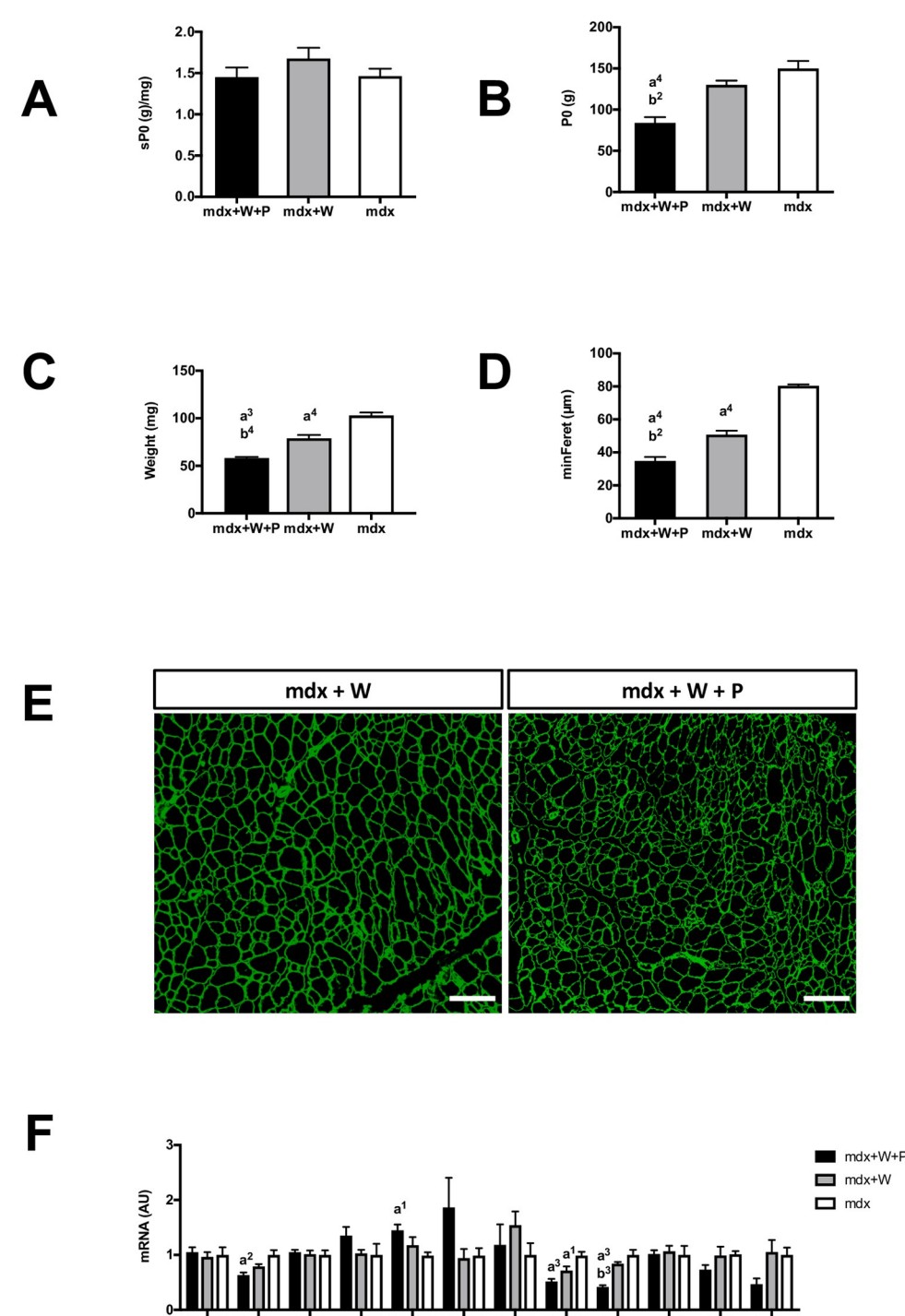

**Fig 4. Effect of *Prox1* transfer on absolute (P0) and specific (sP0) maximal forces, muscle weight and gene expression of atrophy markers in voluntary exercised mdx mice (first set of experiment).** (A) Specific maximal force in Mdx+W+P Mdx+W+P and Mdx+P muscle. n = 6–8 per group. (B) Absolute maximal force in Mdx+W+P and Mdx+P muscle. n = 6–8 per group. (C) Muscle weight in Mdx+W+P and Mdx+P muscle. n = 6–8 per group. (D) Fibre diameters (min feret) in Mdx+W+P and Mdx+P muscle. n = 3–4 per group. (E) Representative image of muscle cross-section. Fiber outline was visualized by antilaminin antibody (green). Scale bar = 200μm. (F) Expression of genes related to atrophy in Mdx+W+P and Mdx+P muscle. N = 6–8 per group. Mdx+W+P: voluntary exercised mdx muscle that received Prox1 transfer into the muscle. Mdx+W: voluntary exercised mdx muscle. Mdx: Mdx muscle. a1, a2, a3, a4: significant different from Mdx, $p < 0.05$, $p < 0.01$, $p < 0.001$, $p < 0.0001$, respectively. b2, b3, b4: significant different from Mdx+W, $p < 0.01$, $p < 0.001$, $p < 0.0001$, respectively.

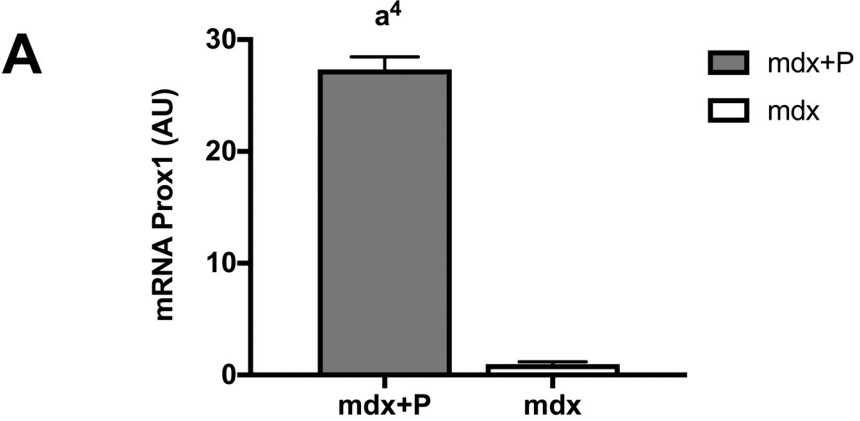

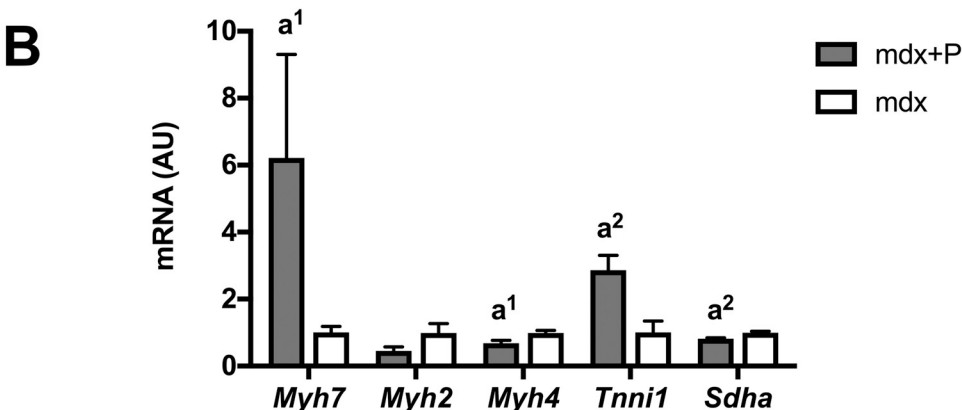

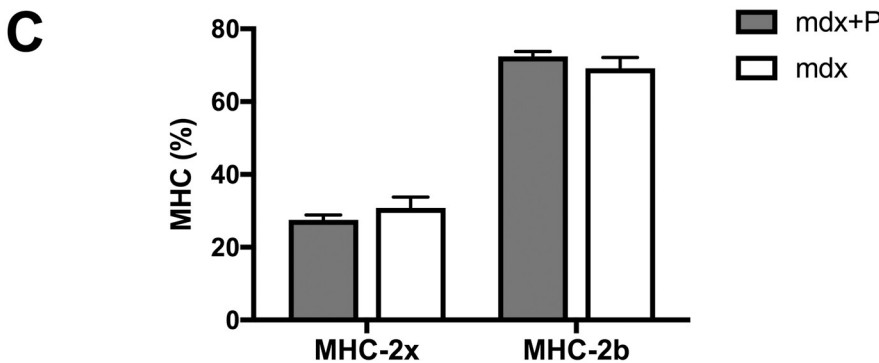

**Fig 5. Effect of *Prox1* transfer on the expression of *Prox1* and markers of fibre type specification in sedentary mdx mice (second set of experiment).** (A) Prox1 expression in Mdx+P and Mdx muscle. N = 6–11 per group. (B) Expression of genes encoding fibre type specific contractile proteins in Mdx+P and mdx muscles. N = 6–11 per group. (C) Relative amounts of MHC-2x and MHC-2b proteins in Mdx+P and Mdx muscle. N = 3 per group. Mdx+P: Mdx muscle that received *Prox1* transfer into the muscle. Mdx: Mdx muscle. a1, a2, a4: significant different from Mdx, $p < 0.05$, $p < 0.01$, $p < 0.0001$, respectively.

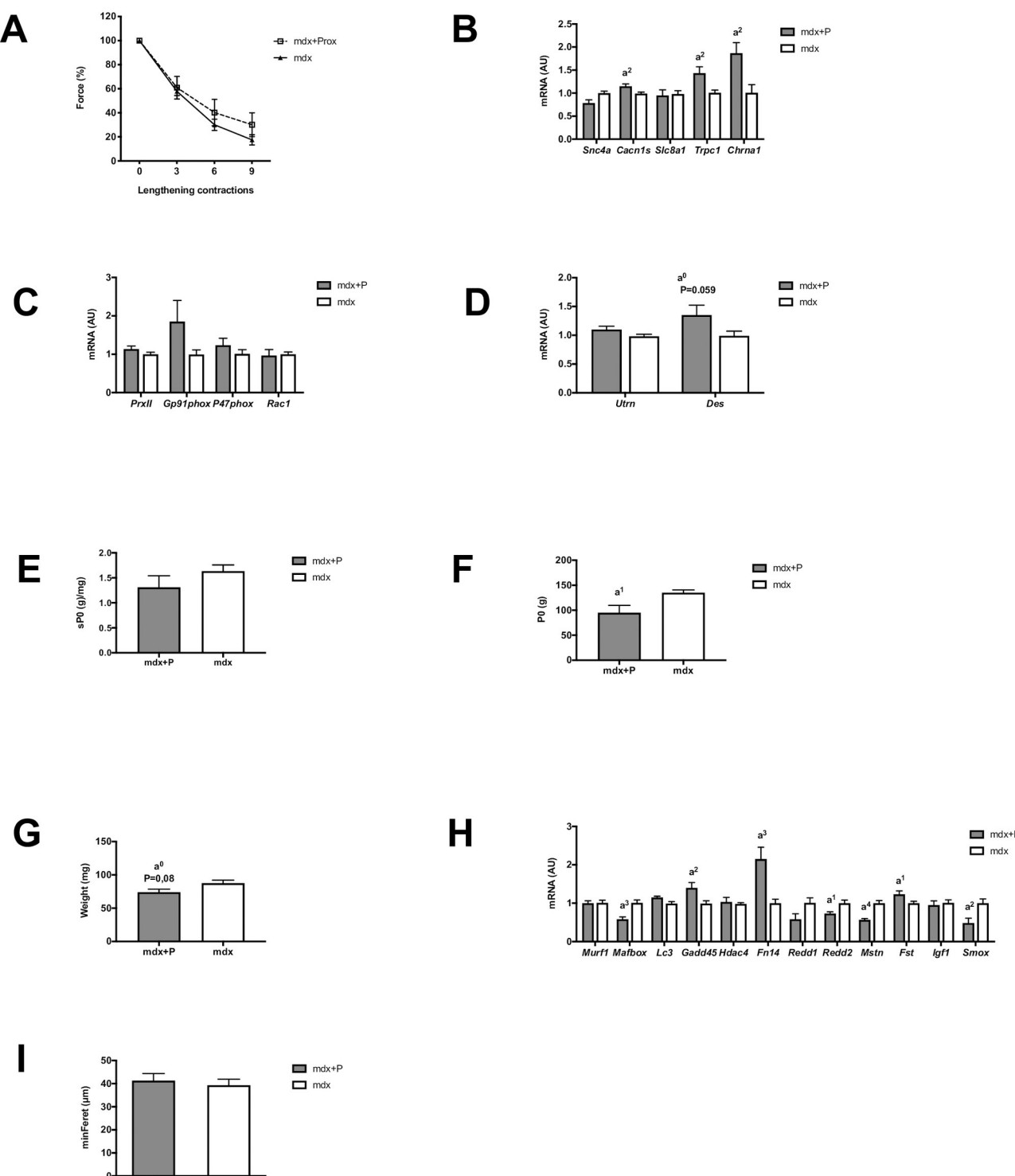

**Fig 6. Effect of *Prox1* transfer on fragility (susceptibility to contraction induced muscle damage) and related gene expression, absolute (P0) and specific (sP0) maximal forces, muscle weight and gene expression of atrophy markers in voluntary exercised mdx mice (first set of experiment) in sedentary mdx mice (second set of experiment).** (A) Force drop following lengthening contractions in Mdx+P and Mdx muscle. n = 5–8 per group. (B) Expression of genes encoding ion channels, related to excitability in Mdx+P and Mdx muscle. N = 6–11 per group. (C) Expression of genes, related to NADPH oxidase 2 (NOX2) in Mdx+P and Mdx muscle. N = 6–11 per group. (D) Expression of genes encoding utrophin (*Utrn*) and desmin (*Des*) in Mdx+P and Mdx muscle. N = 6–11 per group. (E) Specific maximal force in Mdx+P and Mdx muscle. n = 5–8 per group. (F) Absolute maximal force in Mdx+P and Mdx muscle. n = 5–8 per group. (G) Muscle weight in Mdx+P and Mdx muscle. n = 5–9 per group. (H) Expression of genes related to

atrophy in Mdx+P and Mdx muscle. n = 6–11 per group. (I) Fibre diameters (min feret) in Mdx+P and Mdx muscle. n = 3 per group. a1, a2, a3, a4: significantly different from Mdx, p < 0.05, p < 0.01, p < 0.001, p < 0.0001, respectively.

involved in different aspects of muscle function (*Myh7*, *Myh4*, *Tnni1*, *Sdha*, *Trpc1*, *Cacn1s* and *Chrna1*).

### *Prox1* transfer in sedentary Mdx muscle also reduced absolute isometric maximal force

Similarly to voluntary exercised Mdx muscle, we found in the second set of experiment that *Prox1* transfer in Mdx+P muscle did not change specific maximal isometric force (Fig 6E), reduced absolute maximal isometric force (x 0.7)(p < 0.05) (Fig 6F), reduced muscle weight (x 0.8) although not significantly (p = 0.08) (Fig 6G), and decreased the expression of *Mstn* (p < 0.0001) (Fig 6H). Moreover, it decreased absolute maximal lengthening force in Mdx+P muscle (205.9 g ± 18.8) as compared to Mdx muscle (257.0 g ± 15.3), although not significantly (p = 0.07). In contrast to voluntary exercised Mdx muscle, *Prox1* transfer did not change the fibre diameter (Fig 6I). Furthermore, it decreased the expression of *Mafbox* (p < 0.001), *Reed2* (p < 0.05) and *Smox* (p < 0.01), whereas it increased the expression *Gadd45* (p < 0.01), *Fn14* (p < 0.001), and *Fst* (p < 0.05) in Mdx+P muscle (Fig 6H).

## Discussion

### *Prox1* transfer improved fragility in voluntary exercised Mdx mice

The present study confirms previous studies [9, 19] showing that voluntary exercise alleviates the great susceptibility to lengthening contraction-induced force drop, a major dystrophic feature, in fast anterior crural muscles (TA and extensor digitorum longus) of *mdx* mice, such as *Dmd* based preclinical therapy [5]. For the first time, we demonstrate that *Prox1* transfer further improves fragility in voluntary exercised *mdx* mice. Importantly, the muscle was protected from damaging lengthening muscle contractions by *Prox1* transfer only when *mdx* mice performed voluntary exercise. This improved fragility observed in exercised *mdx* mice treated with *Prox1* transfer might be very interesting if it is assumed that fragility causes the exhaustion of the muscle stem cells during successive degeneration/repair cycles [17]. *Prox1* transfer might reduce the progressive muscle wasting in exercised dystrophic muscle because of the promotion of less fragile fibres.

This beneficial effect of *Prox1* transfer in exercised *mdx* mice could be explained by a lower work and stress during lengthening [3]. However, we found that absolute maximal lengthening force (presumably work) is reduced in exercised *mdx* mice, in proportion to the absolute maximal isometric force. We previously observed no strong association between fragility and lengthening force in *mdx* mice, when muscle is maximally activated and for a constant stretch [19]. Indeed, fragility was increased by inactivity and reduced by voluntary exercise in *mdx* mice whereas absolute maximal lengthening force was respectively reduced and unchanged [19].

The reduced fragility induced by *Prox1* transfer in exercised *mdx* mice is associated with the promotion of slower contractile features (increased and reduced expression of *Myh7* and *Myh4* respectively, reduced and increased relative amounts of MHC-2b and MHC-2x proteins respectively). This relation between improved fragility and slower contractile features is in line with the 2 following points. First, slow muscle appears less fragile than fast muscle in *mdx* mice [4, 37]. Second, exercise and pharmacological or genetic activation of signaling pathways, such as calcineurin, PPAR-β, PGC1-α, and AMPK, that promote a slower and more oxidative

gene program, improve fragility in *mdx* mice [9, 18, 19, 23, 27–29, 38]. It was previously demonstrated that *Prox1* promotes slower features, and activates the NFAT-calcineurin pathway [31], a signaling pathway known to play an important role in fibre type specification [39].

It is also possible that *Prox1* transfer improves fragility in voluntary exercised *mdx* mice by a preserved excitability, as voluntary exercise and *Dmd* based therapy [5, 9]. In our experiments, reduced excitability, i.e. plasmalemma electrical dysfunction leading to defective generation and propagation of muscle potential action, largely contributes to the immediate force drop following lengthening contractions in mdx mice [5, 9], in agreement with other studies [7, 8]. Membrane ion channels are likely damaged following by lengthening contractions and *Prox1* transfer possibly interferes with this process. It remains to be determined whether the upregulation of the membrane ion stretch-activated channel *Trpc1* induced by *Prox1* transfer in voluntary exercised *mdx* mice contributes to this improvement of excitability. However, a higher level of TRPC1 or activity of stretch-activated channels are generally associated with a worst dystrophic phenotype and fragility [40, 41]. In line with the present study, it was previously reported that the improved TA *mdx* muscle excitability and fragility induced by voluntary exercise and calcineurin pathway activation were also related to the changes in expression of genes encoding membrane ion channels [9].

Previous studies suggest that increased NOX2 activity is related to fragility in *mdx* mice [13, 14]. However, our results show that *Prox1* transfer in exercised *mdx* muscle does not reduce the expression of *Nox2* subunits (*Gp91phox*, *P47phox* and *Rac1*), which are shown to produce an elevated level of ROS in *mdx* mice [14]. Moreover, we found no increased expression of the gene encoding the antioxidant enzyme *PrxII*, whose overexpression improves fragility in *mdx* mice [13]. Finally, we found that the improvement of the fragility in response to *Prox1* transfer is not associated with significantly increased expression of *Utr* and *Des* in exercised *mdx* mice, two genes contributing to fragility in *mdx* mice [15, 16].

Of note, *Prox1* transfer alone does not significantly improve fragility in sedentary *mdx* mice. It is not excluded that the increase in the number of sedentary *mdx* mice per group change this conclusion but the potential beneficial effect would nevertheless be less important. The difference cannot be attributed to the fact that *Prox1* was not highly overexpressed in sedentary *mdx* mice treated with *Prox1* transfer. However, some changes induced by *Prox1* transfer are notably different between exercised and sedentary *mdx* mice: absolute maximal lengthening force (x 0.6 versus none significant change), MHC-2b (x 0.8 versus none), MHC-2x (x 1.6 versus none), *Myh7* (x 15.1 versus x 6.2), and *Trpc1* (x 2.1 versus x 1.4). Thus, our study interestingly indicates that voluntary exercise potentiates a possible gene-based therapy, at least in the preclinical field.

### *Prox1* transfer reduces maximal force production in exercised *mdx* mice

Although *Prox1* transfer improves fragility in voluntary exercised *mdx* muscle, we found that it has a detrimental effect on absolute maximal isometric force (and maximal lengthening force), without change in specific maximal isometric force. The reduced maximal isometric force is related to a reduced muscle weight and fibre diameter and is associated to the downregulation of *Mstn*, a negative regulator of muscle growth in *mdx* muscle [42]. The same effects were also observed in sedentary *mdx* mice, although less marked. In line with the reduced muscle weight induced by *Prox1* transfer, several genetic or pharmacological treatments promoting slower and more oxidative fibres has been shown to induce muscle atrophy/reduced weight in sedentary *mdx* mice [27, 28, 30], for reasons still largely unknown. It remains to be determined whether the injection of the AAV itself also contributes to the muscle weight reduction (independently of the overexpression of Prox1).

## Conclusions

Combined to voluntary exercise, *Prox1* transfer using an AAV further improves (reduced) the immediate isometric force drop following lengthening contractions. This beneficial effect on fragility in exercised *mdx* mice is associated to the reduction in maximal lengthening force, the promotion of slower contractile features, and the change in *Trpc1* expression. However, *Prox1* transfer also reduces absolute maximal isometric force production. Thus, *Prox1* transfer combined to chronic exercise has effects, some of which are beneficial for the *mdx* dystrophic muscle. Is this knowledge could be exploited for therapeutic advantage?

## Supporting information

**S1 Fig. MHC electrophoresis.**
(PDF)

**S2 Fig. Force record set 1.**
(PDF)

**S3 Fig. Force record set 2.**
(PDF)

**S4 Fig. Fibre diameters set 1.**
(PDF)

**S5 Fig. Fibre diameters set 2.**
(PDF)

**S1 Table. mRNA.**
(PDF)

**S2 Table. MHC electrophoresis.**
(PDF)

**S3 Table. Force and weight.**
(PDF)

**S4 Table. Fibre diameters set 1.**
(PNG)

**S5 Table. Diameters set 2.**
(PNG)

## Acknowledgments

We are grateful to Kari Alitalo (Wihuri Research Institute and Translational Cancer Biology Program, University of Helsinki, Finland) for the gift of Prox1 construct, Pierre Joanne (Sorbonne Université) for assistance during the experiments, Laura Julien and Sofia Benkhelifa for AAV-Prox1 production (Sorbonne Université), Delphine Bouteiller for qPCR measurements (Sorbonne Université), and Saline Jabr (Sorbonne Université) for her help in english.

## Author Contributions

**Conceptualization:** Arnaud Ferry.

**Data curation:** Alexandra Monceau.

**Formal analysis:** Alexandra Monceau, Clément Delacroix, Mégane Lemaitre, Arnaud Ferry.

**Funding acquisition:** Onnik Agbulut.

**Investigation:** Alexandra Monceau, Clément Delacroix, Mégane Lemaitre, Gaelle Revet, Onnik Agbulut, Arnaud Klein, Arnaud Ferry.

**Methodology:** Alexandra Monceau, Mégane Lemaitre, Onnik Agbulut, Arnaud Klein.

**Project administration:** Denis Furling, Onnik Agbulut, Arnaud Ferry.

**Resources:** Denis Furling, Onnik Agbulut.

**Software:** Alexandra Monceau.

**Supervision:** Denis Furling, Onnik Agbulut, Arnaud Klein.

**Validation:** Onnik Agbulut, Arnaud Ferry.

**Visualization:** Alexandra Monceau, Clément Delacroix, Gaelle Revet.

**Writing – original draft:** Arnaud Klein, Arnaud Ferry.

**Writing – review & editing:** Denis Furling, Onnik Agbulut, Arnaud Klein, Arnaud Ferry.

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
