## [Decision Letter · Decision Letter 0]

13 Aug 2021

PONE-D-21-17707

The beneficial effect of chronic muscular exercise on muscle fragility is increased by Prox1 gene transfer in dystrophic mdx muscle

PLOS ONE

Dear Dr. Ferry,

Thank you for submitting your manuscript to PLOS ONE. After careful consideration, we feel that it has merit but does not fully meet PLOS ONE’s publication criteria as it currently stands. Therefore, we invite you to submit a revised version of the manuscript that addresses the points raised during the review process.

We look forward to receiving your revised manuscript.

Kind regards,

Atsushi Asakura, Ph.D

Academic Editor

PLOS ONE

Reviewers' comments:

Reviewer's Responses to Questions

**Comments to the Author**

1. Is the manuscript technically sound, and do the data support the conclusions?

Reviewer #1: Partly

Reviewer #2: Partly

2. Has the statistical analysis been performed appropriately and rigorously? 

Reviewer #1: I Don't Know

Reviewer #2: Yes

3. Have the authors made all data underlying the findings in their manuscript fully available?

Reviewer #1: Yes

Reviewer #2: Yes

4. Is the manuscript presented in an intelligible fashion and written in standard English?

Reviewer #1: Yes

Reviewer #2: Yes

5. Review Comments to the Author

Reviewer #1: In this study, Monceau et al. investigated the effect of Prospero-related homeobox factor

1 gene (Prox1) transfer on fragility in chronically exercised or sedentary mdx mice. The authors concluded that Prox1 transfer reduced the force drop following lengthening contractions in exercised mdx mice, but not in sedentary mdx mice. Whereas, absolute maximal force and muscle weight were reduced by Prox1 transfer, especially in exercised mdx mice. In addition, based on the reduced muscle weight, the authors focused on atrophic genes.

Major concerns

1. The authors concluded the no effect of Prox1 transfer on the fragility and muscle weight in second experiments. In fact, there is no significant difference, but the data show the tendency to decrease. Is there a possibility that addition of mice changes the conclusion? The results of absolute maximal lengthening force (line 393-394) are also not significant, but actually the force was 20% decreased in Mdx+P group.

2. Lines 409-410; The present study confirms previous studies (8,18) showing that voluntary exercise alleviates the great susceptibility to contraction induced injury, a major dystrophic feature, in fast anterior crural muscles (TA and extensor digitorum longus) of mdx mice,

Lines 497-499; Combined to voluntary exercise, Prox1 transfer further improves (reduced) fragility, whereas the single Prox1 transfer approach in sedentary mdx mice failed to induce a

significant effect on the susceptibility to exercise-induced muscle injury.

Is muscle fragility really equal to susceptibility to muscle injury? In this study, there is no data indicating the reduced injury area in Mdx+W+P mice.

3. The authors texted some genes related with muscle atrophy. However, in this study, there is no evidence indicating the decreased myofiber size. The decreased muscle size, but not muscle weight is primary criteria for muscle atrophy.

Minor concerns

1. Line 273; Please correct Figure 1C to Figure 2C.

2. Line 273-276; Immunohistological analyses revealed that these changes were not associated with the modification in the percentages of MHC-1, MHC-2a and MHC-2x expressing fibres because they were not different between Mdx+W+P muscle and Mdx+W muscle (Figures 2D and 2E).

Line 283-284; These data indicate that intramuscular delivery of AAV-283 Prox1 induced a substantial fast to slow contractile transition in the TA muscle of voluntary exercised Mdx mice.

It is unclear the definition of ‘substantial fast to slow contractile transition’ without the no changes of myofiber composition.

3. To reviewer’s knowledge, Prox1 is well known as the marker and regulator for lymphangiogenesis. Reviewer recommends to add some description about the role of Prox1 in lymphangiogenesis.

Reviewer #2: The authors investigated the effects of 1-month voluntary wheel-running exercise and/or Prospero-related homeobox factor 1 gene (Prox1) transfection in dystrophic mdx TA muscle. Prox1 transfection induced a fast-to-slow transition of the myosin isoform and reduced the lengthening-contraction-associated force decline in the TA. One month of running exercise was also found to partially attenuate the lengthening-contraction-associated force decline in the TA. This exercise-associated effect was enhanced by Prox1 transfection. The authors concluded that the Prox1-atransfection combined with voluntary exercise improves the fragility of the mdx muscle. While their results are fairly interesting, this manuscript has several concerns to address.

The authors report that the muscle weight was decreased by Prox1 transfection and/or 1-month voluntary wheel running. The authors should report the body weight, together with the muscle weight relative to the body weight, of each experimental group. These are important physiological parameters to evaluate the effects of Prox1 transfection and exercise.

While the muscle weight was decreased by Prox1 transfection and/or voluntary running, no significant difference was found in the mean fiber diameters of the respective fiber types among the experimental groups. What explains these discrepant results? Did these treatments decrease the number of muscle fibers?

Prox1 transfection and/or voluntary exercise partially attenuated the lengthening-contraction-associated force decline in the TA. The relative reduction may be inversely related with the level of absolute maximal force. Do these results suggest that the lower level of maximal tension elicits the beneficial effects of Prox1 transfection and voluntary exercise on the fragility of the mdx TA muscle? Though the authors discuss this issue, they should take the added step of offering a clear explanation. The amplitude of the absolute maximal force had a large impact on the relative percentage of force reduction. Therefore, the authors should also evaluate the lengthening-associated force reduction using the absolute force.

Figure 2

An image of the mdx muscle should be provided.

Judging from the images, the combination of Prox1 and voluntary exercise drastically increased the population of type 2x fibers, and decreased the populations of type 2a and 2b fibers, compared to the exercised-alone group. These changes are somewhat smaller than those shown in Figure 2C. What explains the difference?

All of the evaluations in this study were carried out using the TA muscle. The fiber type distribution of the TA muscle is region-specific. The authors state that they counted all fibers in a cross-section of the mid-belly region of the TA. Figure 2D, on the other hand, shows only a limited portion of the TA muscle.

The cause of the lengthening-associated force reduction remains unclear. The intracellular Ca2+ level has an impact on the force generation. The authors should provide a record of the force curve with a baseline level (namely, the changes in the resting tension during successive contractions). Further, they should evaluate the molecules that handle intracellular Ca2+, such as those associated with the functions　sarcoplasmic reticulum (ryanodine receptor, Ca2+ pump), T-tubes (DHP receptor, caveolin), parvalbumin, and Na+-Ca2+ exchanger.

6. PLOS authors have the option to publish the peer review history of their article (what does this mean?). If published, this will include your full peer review and any attached files.

Reviewer #1: No

Reviewer #2: No

---

## [Author Response · Author response to Decision Letter 0]

20 Sep 2021

Response to Reviewers

Academic Editor

• A rebuttal letter that responds to each point raised by the academic editor and reviewer(s). You should upload this letter as a separate file labeled 'Response to Reviewers'. AUTHORS’RESPONSE:This was done.

• A marked-up copy of your manuscript that highlights changes made to the original version. You should upload this as a separate file labeled 'Revised Manuscript with Track Changes'. AUTHORS’RESPONSE: This was done.

• An unmarked version of your revised paper without tracked changes. You should upload this as a separate file labeled 'Manuscript'. AUTHORS’RESPONSE: This was done.

Journal requirements

AUTHORS’RESPONSE: We checked Plos One’s style requirements.

AUTHORS’RESPONSE: Supporting information files concerning minimal data set were uploaded. 

Reviewer #1:

 In this study, Monceau et al. investigated the effect of Prospero-related homeobox factor

1 gene (Prox1) transfer on fragility in chronically exercised or sedentary mdx mice. The authors concluded that Prox1 transfer reduced the force drop following lengthening contractions in exercised mdx mice, but not in sedentary mdx mice. Whereas, absolute maximal force and muscle weight were reduced by Prox1 transfer, especially in exercised mdx mice. In addition, based on the reduced muscle weight, the authors focused on atrophic genes.

Major concerns

1. The authors concluded the no effect of Prox1 transfer on the fragility and muscle weight in second experiments. In fact, there is no significant difference, but the data show the tendency to decrease. Is there a possibility that addition of mice changes the conclusion? The results of absolute maximal lengthening force (line 393-394) are also not significant, but actually the force was 20% decreased in Mdx+P group. AUTHORS’RESPONSE: Yes, it is possible. We have added this notion in the Discussion.

2. Lines 409-410; The present study confirms previous studies (8,18) showing that voluntary exercise alleviates the great susceptibility to contraction induced injury, a major dystrophic feature, in fast anterior crural muscles (TA and extensor digitorum longus) of mdx mice,

Lines 497-499; Combined to voluntary exercise, Prox1 transfer further improves (reduced) fragility, whereas the single Prox1 transfer approach in sedentary mdx mice failed to induce a

significant effect on the susceptibility to exercise-induced muscle injury.

Is muscle fragility really equal to susceptibility to muscle injury? In this study, there is no data indicating the reduced injury area in Mdx+W+P mice. AUTHORS’RESPONSE: The drop in maximal force following eccentric contractions is a marker recognized as relevant, emerging as a robust phenotype of murine dystrophy. This notion is underlined in the Introduction. It may give information different from that of histological markers. Indeed, it is possible to found a dramatic drop in maximal force for a few days although histological lesions are only minor.

3. The authors texted some genes related with muscle atrophy. However, in this study, there is no evidence indicating the decreased myofiber size. The decreased muscle size, but not muscle weight is primary criteria for muscle atrophy. AUTHORS’RESPONSE: This notion was added in the Discussion. The weight of the muscle is related to the volume of the muscle (weight = volume x density).

Minor concerns

1. Line 273; Please correct Figure 1C to Figure 2C. AUTHORS’RESPONSE: This is now revised.

2. Line 273-276; Immunohistological analyses revealed that these changes were not associated with the modification in the percentages of MHC-1, MHC-2a and MHC-2x expressing fibres because they were not different between Mdx+W+P muscle and Mdx+W muscle (Figures 2D and 2E).

Line 283-284; These data indicate that intramuscular delivery of AAV-283 Prox1 induced a substantial fast to slow contractile transition in the TA muscle of voluntary exercised Mdx mice.

It is unclear the definition of ‘substantial fast to slow contractile transition’ without the no changes of myofiber composition. AUTHORS’RESPONSE: The notion of a substantial rapid to slow transition is now revised.

3. To reviewer’s knowledge, Prox1 is well known as the marker and regulator for lymphangiogenesis. Reviewer recommends to add some description about the role of Prox1 in lymphangiogenesis. AUTHORS’RESPONSE: The importance of Prox1 for lymphatic vessels is now added (Introduction).

Reviewer #2: 

The authors investigated the effects of 1-month voluntary wheel-running exercise and/or Prospero-related homeobox factor 1 gene (Prox1) transfection in dystrophic mdx TA muscle. Prox1 transfection induced a fast-to-slow transition of the myosin isoform and reduced the lengthening-contraction-associated force decline in the TA. One month of running exercise was also found to partially attenuate the lengthening-contraction-associated force decline in the TA. This exercise-associated effect was enhanced by Prox1 transfection. The authors concluded that the Prox1-atransfection combined with voluntary exercise improves the fragility of the mdx muscle. While their results are fairly interesting, this manuscript has several concerns to address.

The authors report that the muscle weight was decreased by Prox1 transfection and/or 1-month voluntary wheel running. The authors should report the body weight, together with the muscle weight relative to the body weight, of each experimental group. These are important physiological parameters to evaluate the effects of Prox1 transfection and exercise. AUTHORS’RESPONSE: One leg is injected with Prox1 while the other leg is injected with PBS (Figure 1). One leg is injected with Prox1 while the other leg of the same mouse is injected with PBS (Figure 1). The body weights of the mice are therefore not different between the M + W + P and M + W groups. Ditto for groups M + P and M (Figure 1). Body weights is therefore not an interesting measured parameter. Body weights corresponding to M+W+P and M+W muscles were approximatively 30 g (28.g to 30.5 g), this was added in Materials and Methods.

While the muscle weight was decreased by Prox1 transfection and/or voluntary running, no significant difference was found in the mean fiber diameters of the respective fiber types among the experimental groups. What explains these discrepant results? Did these treatments decrease the number of muscle fibers? AUTHORS’RESPONSE: One possibility is that the number of fibers is decreased. Unfortunately, we could not measure this parameter with much precision because of the unsatisfactory quality of some cross sections of the muscle (part of the cross sections). Anyway, the possibility of a decrease in the number of fibers in 4 weeks is low because this type of phenomenon has not yet been described in previous studies. A second possibility is that the density of the muscle has changed (weight = volume x density). This latter notion is now added in the revised Discussion).

Prox1 transfection and/or voluntary exercise partially attenuated the lengthening-contraction-associated force decline in the TA. The relative reduction may be inversely related with the level of absolute maximal force. Do these results suggest that the lower level of maximal tension elicits the beneficial effects of Prox1 transfection and voluntary exercise on the fragility of the mdx TA muscle? Though the authors discuss this issue, they should take the added step of offering a clear explanation. The amplitude of the absolute maximal force had a large impact on the relative percentage of force reduction. Therefore, the authors should also evaluate the lengthening-associated force reduction using the absolute force. AUTHORS’RESPONSE: AUTHORS’RESPONSE: We added a clearer explanation in Discussion as suggested. Results concerning the reduction of maximal force are already presented in Results: “Similarly, absolute maximal lengthening force was lower (x 0.6) in Mdx+W+P muscle (157.2 g ± 7.5) compared to Mdx+W muscle (240.0 g ± 10.8) muscle (p < 0.01). In addition, the ratio of absolute maximal lengthening force to the absolute maximal isometric force was not different between Mdx+W+P muscle (1.9 ± 0.1) and Mdx+W muscle (1.8 ±329 0.1).”

Figure 2

An image of the mdx muscle should be provided. AUTHORS’RESPONSE: This is now added in supporting information files. We added all the images of muscles. 

Judging from the images, the combination of Prox1 and voluntary exercise drastically increased the population of type 2x fibers, and decreased the populations of type 2a and 2b fibers, compared to the exercised-alone group. These changes are somewhat smaller than those shown in Figure 2C. What explains the difference? AUTHORS’RESPONSE: 1) When we measured all the images using immunohistochemistry, there are no differences in the number of fibers expressing neither MHC-2b nor MHC-2a between Mdx+W+P muscle and Mdx+W muscle, as shown in Figure 2E. We also found no significant difference between these two groups concerning the number of fibres expressing MHC-2x. 2) The variability between muscles and portions of muscle can explained the difference between Figure 2E and Figure 2D. So, based on the reviewer's remarks, we decided to delete Figure 2D. All the images of the muscles are now shown in supporting information files. 2) The discrepancy between the Figure 2C and Figure 2E could be explained by the fact that Figure 2C considers the co-expression of MHC, unlike Figure 2E. A second explanation could come from the fact that Figure 2C is more quantitative while Figure 2E is more qualitative. In the revised manuscript, we modified the notion of a substantial fast to slow transition since the effect of Prox1 on the populations on the different types of fibres is modest compared to what expected, and more complicated. 

All of the evaluations in this study were carried out using the TA muscle. The fiber type distribution of the TA muscle is region-specific. The authors state that they counted all fibers in a cross-section of the mid-belly region of the TA. Figure 2D, on the other hand, shows only a limited portion of the TA muscle. AUTHORS’RESPONSE: We agree with the reviewer. We deleted the Figure 2D. The full images are now shown in supporting information files. 

The cause of the lengthening-associated force reduction remains unclear. The intracellular Ca2+ level has an impact on the force generation. The authors should provide a record of the force curve with a baseline level (namely, the changes in the resting tension during successive contractions). AUTHORS’RESPONSE: We found no change in the baseline level. Force records are now shown supporting information files. 

Further, they should evaluate the molecules that handle intracellular Ca2+, such as those associated with the functions　sarcoplasmic reticulum (ryanodine receptor, Ca2+ pump), T-tubes (DHP receptor, caveolin), parvalbumin, and Na+-Ca2+ exchanger. AUTHORS’RESPONSE: We agree with the reviewer, the study of the functions of the proteins involved in excitation would be very interesting, but this goes well beyond the scope of this study. Previous studies (cited in the manuscript) show that treatments (gene therapy aimed at restoring dystrophin expression, voluntary exercise) that improve muscle fragility in mdx mice improve muscle excitability, through mechanisms not still well known

While revising your submission, please upload your figure files to the Preflight Analysis and Conversion Engine (PACE) digital diagnostic tool, https://pacev2.apexcovantage.com/. PACE helps ensure that figures meet PLOS requirements. To use PACE, you must first register as a user. Registration is free. Then, login and navigate to the UPLOAD tab, where you will find detailed instructions on how to use the tool. If you encounter any issues or have any questions when using PACE, please email PLOS at figures@plos.org. Please note that Supporting Information files do not need this step. AUTHORS’RESPONSE: The new Fig2 was uploaded to the PACE tool.

---

## [Decision Letter · Decision Letter 1]

6 Dec 2021

PONE-D-21-17707R1The beneficial effect of chronic muscular exercise on muscle fragility is increased by Prox1 gene transfer in dystrophic mdx musclePLOS ONE

Dear Dr. Ferry,

Thank you for submitting your manuscript to PLOS ONE. After careful consideration, we feel that it has merit but does not fully meet PLOS ONE’s publication criteria as it currently stands. Therefore, we invite you to submit a revised version of the manuscript that addresses the points raised during the review process as below.

We identified substantive lack of responses to reviewer's requests, especially concerned about the conclusions which cannot be justified on the basis of the current form of the manuscript. The editors concur. Given the nature of these concerns, we regret that we cannot accept the manuscript. 

We look forward to receiving your revised manuscript.

Kind regards,

Atsushi Asakura, Ph.D

Academic Editor

PLOS ONE

Reviewers' comments:

Reviewer's Responses to Questions

**Comments to the Author**

1. If the authors have adequately addressed your comments raised in a previous round of review and you feel that this manuscript is now acceptable for publication, you may indicate that here to bypass the “Comments to the Author” section, enter your conflict of interest statement in the “Confidential to Editor” section, and submit your "Accept" recommendation.

Reviewer #1: (No Response)

Reviewer #2: (No Response)

2. Is the manuscript technically sound, and do the data support the conclusions?

Reviewer #1: No

Reviewer #2: No

3. Has the statistical analysis been performed appropriately and rigorously? 

Reviewer #1: I Don't Know

Reviewer #2: Yes

4. Have the authors made all data underlying the findings in their manuscript fully available?

Reviewer #1: (No Response)

Reviewer #2: Yes

5. Is the manuscript presented in an intelligible fashion and written in standard English?

Reviewer #1: (No Response)

Reviewer #2: Yes

6. Review Comments to the Author

Reviewer #1: Related to major comment #1

Authors’ response; Line 462; It is nevertheless not excluded that the addition of mice changes this conclusion.

Additional experiment should be performed to confirm the possibility.

Related to major comment #2

Reviewer’s question means that the impact of Prox1 on susceptibility to muscle injury, because the authors described this concerns in line 409-410 (revised 403-404) and 497-499 (revised 492-494). Without the results of histological analyses, the authors can not mention the impact of Prox1 on susceptibility to muscle injury. The description in lines 492-494 also needs to be correct as sedentary group did not exercise.

Related to major comment #3

Reviewer disagrees with the authors’ response because number of myofiber also affect the muscle wight. As there is not difference in the data of diameter, the data of myofiber size should be included.

Reviewer #2: The authors made a few slight modifications to their manuscript in response to my previous comments. Regrettably, their responses and revisions do not go far enough. While the authors added supplemental data, their other revisions were minimal. The quality of the cross-sectional images of the muscle is a particular concern.

The authors’ responses and explanations on the discrepancy between their results on the muscle weights and fiber CSAs were insufficient. Both increases in number of muscle fibers and density of muscle fibers are unlikely. This finding can be attributed to technical errors during the fixation, the sectioning, and/or the staining procedures. Judging from the supplemental data (S4-S9), the very low quality of the images prevented the authors from accurately determining the fiber CSAs. These poor images reflect the low quality of the data in this study overall.

As mentioned, the quality of the immuno-stained images is too low. As a reviewer, I am unable to judge the MHC type of each fiber. I believe that types 2a and 2b are co-expressed in many types of fibers. The authors should carry out staining on new sections and then show not only merged images but also separate images stained with each type of MyHC.

Figures 3 and 4:

The authors should show the body weights of the mdx mice. They should also evaluate the difference in body weights between the mdx group and Prox1-transferred mdx group with or without exercise. Why, moreover, did the muscle weight decrease by ~25% following the Prox1 gene transfer only during the 4-week experimental period? And why did the weights of the mdx mice with both Prox1 and exercise decrease by ~50% compared to the mdx control during the 4-week experimental period? Was this reduction of muscle weight attributable to atrophy? Were physiological factors responsible for the abnormal changes in muscle weight induced by Prox1?

7. PLOS authors have the option to publish the peer review history of their article (what does this mean?). If published, this will include your full peer review and any attached files.

Reviewer #1: No

Reviewer #2: No

---

## [Author Response · Author response to Decision Letter 1]

7 Jan 2022

1) Responses to Editor comments:

This manuscript is a revised version of the manuscript : PONE-D-21-17707

The beneficial effect of chronic muscular exercise on muscle fragility is increased by Prox1 gene transfer in dystrophic mdx muscle. We have previously shown that chronic exercise improves the fragility of a murine dystrophic muscle. In this study, we wanted to determine if the transfer of Prox1 into a exercised dystrophic muscle causes further improved fragility.

We have changed the manuscript according to most of the suggestions of reviewers. We paid particular attention to the conclusions, as requested. Unfortunately, it is not possible to do a third set of experimentation as reviewer #1 would like, with so many muscles in each group at the same time (to avoid experimental bias). In our opinion, this does not diminish much in the interest of our study because the main objective was not to study in detail the effect of the transfer of Prox1 in sedentary mdx mice (since we mainly study the effect of Prox1 transfer on muscle fragility in exercised mdx mice). Likewise, the main objective is not to determine the mechanisms which are responsible for the reduction in muscle weight observed in exercised mdx mice treated with Prox1 transfer, as reviewer # 2 would have liked. 

For us, what is important in the results of this preclinical study is that a transfer of the Prox1 gene via AAV improves the fragility of an exercised dystrophic muscle but also decreases the absolute maximal force, i.e., increases muscle weakness. Thus, the potential clinical benefit of the transfer of Prox1 into murine exercised dystrophic muscle is real but more questionable than expected.

We also revised the Funding statement: 'The author(s) received no specific funding for this work."

We hope that all the attention we have given to the corrections will be considered satisfactory.

Very best regards

Pr. Arnaud Ferry

2) Responses to Reviewers comments: see below.

Reviewer #1: 

Related to major comment #1

Authors’ response; Line 462; It is nevertheless not excluded that the addition of mice changes this conclusion. 

AUTHOR RESPONSE. Major comment #1. Previously, to consider the comment of the reviewer we had indeed added in the Discussion this :“It is nevertheless not excluded that the addition of mice changes this conclusion”. In the revised manuscript we added: “but the potential beneficial effect would nevertheless be less important” (see below).

Additional experiment should be performed to confirm the possibility. 

AUTHOR RESPONSE. “Additional experiment should be performed to confirm the possibility”. The principal purpose of the present study was to determine whether Prox1 transfer, i.e., Prox1 transfer, reduced muscle fragility in voluntary exercised mdx mice, since voluntary running only partly reduced the susceptibility to exercise-induced muscle injury, so it would be interesting to combined the effects of exercise with those of another treatment, i.e., Prox1 transfer. Thus, the main aim is not to study the effect of the transfer of Prox1 in sedentary mdx mice. This study already includes 2 sets of experiments. In our opinion, these 2 sets of experiments already bring important preclinical results in exercised mdx mice, which meets our objectives. However, we agree that it may be of interest in the future to better study the effect of Prox1 transfer in mdx mice, although the effect appears to be more modest. In fact, the p-value is high (p = 0.3989) in the second set of experiment. In order to find a significant difference (p = 0.05) between the 2 groups (sedentary mice treated or not with Prox1), we estimate (calculate) the number of muscles in each of the 2 groups to be more than 20. Unfortunately, it not possible to perform and analyze a such study. It is not possible to do such a third set of experiment, with so many muscles in each group at the same time. Indeed, it would take too much production of mice (only one sex is studied) and viral vector. In addition, the difference of 8.623% between the 2 groups (Mdx + P and Mdx) is small, of no great physiological consequence, even if it had to be found statistically significant using 40 more muscles. 

Related to major comment #2

Reviewer’s question means that the impact of Prox1 on susceptibility to muscle injury, because the authors described this concerns in line 409-410 (revised 403-404) and 497-499 (revised 492-494). Without the results of histological analyses, the authors can not mention the impact of Prox1 on susceptibility to muscle injury. The description in lines 492-494 also needs to be correct as sedentary group did not exercise.

AUTHOR RESPONSE: “Muscle injury”. The immediate drop force following lengthening contractions we used is a very widely accepted way to determine the susceptibility to exercise-induced muscle damage (see Introduction). Below are three examples: 1) “Lack of dystrophin renders skeletal muscle susceptible to injury, particularly eccentric contraction (ECC)-induced strength loss.” (Skeletal Muscle. 2020; 10: 3.), 2) “Susceptibility to ECC contraction-induced strength loss has therefore become a standard outcome measure in preclinical studies to assess disease severity and the efficacy of potential therapies for DMD.” (Med Sci Sports Exerc. 2020 Feb;52(2):354-361) and 3) “Whether whole muscles are studied in vitro, in situ, or in vivo, the overwhelming evidence indicates that whole skeletal muscles of mdx mice show a greater susceptibility to contraction-induced injury than muscles of control mice. The most compelling data indicate a 20% greater force deficit for EDL muscles of mdx mice compared with those of control mice with both studied in situ.”(Am J Physiol Cell Physiol. 2000 Oct;279(4):C1290-4.). It is this notion which is defined at the beginning of the Introduction of our manuscript, and which we use in our manuscript. Moreover, in a recent study, we found no histological structural change following immediate lengthening contractions in mdx mice (Roy et al 2016, Skeletal Muscle. 2016 Jul 20;6:23.). This last notion was also added in the revised manuscript (Introduction), in order to better define what fragility is. Moreover, we have considered the reviewer's remark in our revised manuscript (Abstract, Discussion, conclusion…). In particular, we have removed the notion of injury.

AUTHOR RESPONSE. “The description in lines 492-494 also needs to be correct”. Mdx mice (sedentary Mdx mice) did not perform chronic voluntary running (see Material and Methods) but their TA muscles performed 9 lengthening contractions. Please, but do not confuse the single session of 9 lengthening contractions in response to nerve electric stimulation with chronic voluntary running in a wheel. The 9 lengthening contractions are performed to evaluate the “Susceptibility to ECC contraction-induced strength loss”, fragility.

Related to major comment #3

Reviewer disagrees with the authors’ response because number of myofiber also affect the muscle wight. As there is not difference in the data of diameter, the data of myofiber size should be included.

AUTHOR RESPONSE. “the data of myofiber size should be included”. The data of myofiber size were already included in the manuscript R-1 (Figures 4D and 6H). However, the data of diameter are now deleted in the R-2 manuscript because Reviewer #2 have criticisms regarding the quality of the immunostained image. This is the reason why we prefer to not show data concerning the number of fibers in the previous version of the manuscript, which requires a perfect quality of muscle cross section images, which is difficult to obtain in dystrophic mice with many muscle tissue alterations (very small fibres and cells, splitted fibers, branched fibers…), in addition to the absence of dystrophin which did not allow a good staining of the contours of the fibers. For example, half of the fibers were branched in young adult mdx mice (Front Physiol. 2021 Dec 7;12:771499.). Thus, we deleted all histological data in the revised manuscript. This is not too regrettable as the effect of the transfer of Prox1 on fibre size/number is clearly not the main objective of the study. The main objective is to study the effect of the transfer of Prox1 on the muscle fragility of exercised mdx mice. In order to improve this understanding, we have focused less on muscle weight reduction in the revised manuscript.

Reviewer #2: 

The authors made a few slight modifications to their manuscript in response to my previous comments. Regrettably, their responses and revisions do not go far enough. While the authors added supplemental data, their other revisions were minimal. The quality of the cross-sectional images of the muscle is a particular concern.

The authors’ responses and explanations on the discrepancy between their results on the muscle weights and fiber CSAs were insufficient. Both increases in number of muscle fibers and density of muscle fibers are unlikely. This finding can be attributed to technical errors during the fixation, the sectioning, and/or the staining procedures. Judging from the supplemental data (S4-S9), the very low quality of the images prevented the authors from accurately determining the fiber CSAs. These poor images reflect the low quality of the data in this study overall.

AUTHOR RESPONSE. “the very low quality of the images prevented the authors from accurately determining the fiber CSAs”. We agree with the reviewer, the quality of the immune-stained images is not good, probably due to the freezing/preservation of the muscles and structural alterations in dystrophic muscle. We regularly encounter difficulties to obtain nice images in dystrophic mice with many muscle tissue alterations (very small fibres and cells, splitted fibers, branched fibers, fibrosis…), in addition to the absence of dystrophin which did not allow a good staining of the contours of the fibers. For example, half of the fibers were branched in young adult mdx mice (Front Physiol. 2021 Dec 7;12:771499.). As suggested by the reviewer, we deleted all histological data. This is not too regrettable as the effect of the transfer of Prox1 on fibre size/number is clearly not the main objective of the study. The main objective is to study the effect of the transfer of Prox1 on the muscle fragility of exercised mdx mice. In order to improve this understanding, we have focused less on muscle weight reduction in the revised manuscript.

As mentioned, the quality of the immuno-stained images is too low. As a reviewer, I am unable to judge the MHC type of each fiber. I believe that types 2a and 2b are co-expressed in many types of fibers. The authors should carry out staining on new sections and then show not only merged images but also separate images stained with each type of MyHC.

AUTHOR RESPONSE.” I believe that types 2a and 2b are co-expressed in many types of fibers”. We found that very few MHC-2a and MHC-2b are co-expressed (Mdx: 0-10 fibres per TA muscle cross-section; Mdx+W: 1-18 fibres; Mdx+W+P: 2-3 fibres). We also made separate images. To take this remark (low quality of the image) into account, we have deleted all data concerning immunohistology. This is not harmful because we have other markers of fast to slow conversion of fiber type.

Figures 3 and 4:

The authors should show the body weights of the mdx mice. They should also evaluate the difference in body weights between the mdx group and Prox1-transferred mdx group with or without exercise.

AUTHOR RESPONSE. “Difference in body weights”. The range of the body weights of mdx mice (set 1) were already shown. We now shown the mean values � SEM of the 2 groups (29.9 � 0.321 versus 31.4 � 0.23 g, see Materials and methods). There a 4.9% difference in body weight between the 2 groups (mdx group and Prox1-transferred mdx group with or without exercise). The mdx group was not intramuscularly injected, did not voluntary run and was studied 3-4 weeks later (The mdx group was studied later, due to sanitary confinement).

Why, moreover, did the muscle weight decrease by ~25% following the Prox1 gene transfer only during the 4-week experimental period? And why did the weights of the mdx mice with both Prox1 and exercise decrease by ~50% compared to the mdx control during the 4-week experimental period? Was this reduction of muscle weight attributable to atrophy? Were physiological factors responsible for the abnormal changes in muscle weight induced by Prox1? 

AUTHOR RESPONSE. “Why, moreover, did the muscle weight decrease by 25%”. As described in the experimental design (Figure 1), the muscles treated or not with Prox1 come from the same mice. Therefore, they were subjected to the same physiological factors. The only difference is the injection of AAV-Prox1. Either it is Prox1 itself which is responsible for the reduction in muscle weight, or it is the injection of the AAV itself (independent of the overexpression of Prox1). This was now indicated in the text of the revised manuscript (see Discussion). What we do know is that a decrease in muscle weight of sedentary Mdx is also observed following various treatments inducing a change in the type of fibers (see Discussion). We agree with the reviewer that it would be interesting to better analyze the effects of Prox1 transfer on muscle weight. Unfortunately, we have no clear explanation concerning the reduction in the weight of the muscles treated with AAV-Prox1. However, for us, what is important in the results of this preclinical study, with regard to the objective of this study, is that a transfer of the Prox1 gene improves the fragility of the exercised dystrophic muscle, but also decreases the absolute maximal force. Thus, the potential clinical benefit of the transfer of Prox1 into exercised dystrophic muscle is real but more questionable than expected.

AUTHOR RESPONSE. “why did the weights of the mdx mice with both Prox1 and exercise decrease by ~50%”. Concerning the reduced muscle weight of Mdx+W+P compared to Mdx mice we noted these differences between the 2 groups of mice: 1) the Mdx group was not intramuscularly injected with AAV-Prox1, 2) Mdx mice did not voluntary run and 3) Mdx mice were studied 3-4 weeks later (the Mdx group was studied later, due to sanitary confinement). That makes a lot of differences that make a possible explanation even more complicated.

---

## [Editor Report · Decision Letter 2]

11 Feb 2022

PONE-D-21-17707R2The beneficial effect of chronic muscular exercise on muscle fragility is increased by Prox1 gene transfer in dystrophic mdx musclePLOS ONE

Dear Dr. Ferry,

Thank you for submitting your manuscript to PLOS ONE. After careful consideration, we feel that it has merit but does not fully meet PLOS ONE’s publication criteria as it currently stands. Therefore, we invite you to submit a revised version of the manuscript that addresses the points raised during the review process.

While the focus of this study is for functional improvement of mdx muscle after injection of Prox1 AAV vectors, both reviews requested revised histological figures as well as fiber CSA measurements, both of which data would strengthen the mdx muscle phenotype after Prox1 gene transduction. Therefore, please add revised histological figures and perform the fiber CSA measurements,

We look forward to receiving your revised manuscript.

Kind regards,

Atsushi Asakura, Ph.D

Academic Editor

PLOS ONE

---

## [Author Response · Author response to Decision Letter 2]

25 Mar 2022

We have changed the manuscript according to the suggestions. We selected good quality cross-section images of the muscles, and performed new histological measurements which are now shown.

We hope that all the attention we have given to the corrections will be considered satisfactory.

Very best regards

Pr. Arnaud Ferry

---

## [Editor Report · Decision Letter 3]

6 Apr 2022

The beneficial effect of chronic muscular exercise on muscle fragility is increased by Prox1 gene transfer in dystrophic mdx muscle

PONE-D-21-17707R3

Dear Dr. Ferry,

We’re pleased to inform you that your manuscript has been judged scientifically suitable for publication and will be formally accepted for publication once it meets all outstanding technical requirements.

Kind regards,

Atsushi Asakura, Ph.D

Academic Editor

PLOS ONE
---

## [Editor Report · Acceptance letter]

8 Apr 2022

PONE-D-21-17707R3 

The beneficial effect of chronic muscular exercise on muscle fragility is increased by *Prox1* gene transfer in dystrophic *mdx* muscle 

Dear Dr. Ferry:

I'm pleased to inform you that your manuscript has been deemed suitable for publication in PLOS ONE. Congratulations! Your manuscript is now with our production department. 

Kind regards, 

on behalf of

Dr. Atsushi Asakura 

Academic Editor

PLOS ONE